

**PeerJ Hubs**
Published on behalf of

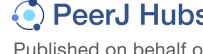
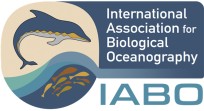

International Association for Biological Oceanography
IABO

# Barnacle analysis as a microplastic pollution bioindicator on the East Coast of Surabaya

Miftakhul Sefti Raufanda[1], Aunurohim Aunurohim[2] and Romanus Edy Prabowo[1]

[1] Faculty of Biology, Universitas Jenderal Soedirman, Banyumas, Indonesia
[2] Department of Biology, Institut Teknologi Sepuluh Nopember, Surabaya, Indonesia

## ABSTRACT

**Background.** Plastic pollution is a significant issue on the East Coast of Surabaya, emphasizing the need to develop microplastic monitoring programs. Barnacles became one of the potential microplastic bioindicator species on the East Coast of Surabaya. This study aimed to characterize the visual and polymers of microplastics found in barnacles and assess their potential as a bioindicator species for microplastic pollution on the East Coast of Surabaya.

**Methods.** Microplastic polymer analysis was performed using ATR-FTIR.

**Results.** A total of 196 microplastic particles were found in barnacles, water, and sediment. The size of microplastics in barnacles, water, and sediment varied, with the size in barnacles dominated by class 1 (1–10 μm), in water by class 2 (10–50 μm), and in sediments by class 3 (50–100 μm). Fragments dominated the shape of microplastics in barnacles, while water and sediment were dominated by fiber. The microplastic color in barnacles, water, and sediment was dominated by blue, and the microplastic polymer composition on barnacles, water, and sediments was dominated by cellophane (36%). *Amphibalanus amphitrite* was found to be predominant and identified as a potential microplastic bioindicator because it is a cosmopolitan species. Its population was found to correlate positively with cellophane (CP) accumulation. The Pearson's correlation test between barnacle length and microplastic length at a = 0.05 was inversely proportional to $r = -0.411$ ($p < 0.05$), categorized as a strong enough correlation. These findings are essential in developing monitoring programs and mitigating the impact of microplastics on the marine environment.

# INTRODUCTION

The issue of plastic pollution in marine waters has gained significant attention from the scientific community and the public due to its potential dangers. About 10% of plastic produced ends up in the oceans, where it accumulates and does not decompose (*Saeed et al., 2020*). Indonesia is considered the second-largest plastic waste contributor globally, with an estimated 0.48−1.29 million metric tons of plastic waste in the sea per year, following China

Corresponding author
Aunurohim Aunurohim,
aunurohim@its.ac.id

(*Jambeck et al., 2015*; *Kurniadi & Hizasalasi, 2017*). The breakdown of plastic into smaller fragments occurs when exposed to ultraviolet radiation from the sun, surface waves, and turbulence of water currents (*Li et al., 2019*; *Saeed et al., 2020*). These small fragments with dimensions less than 5 mm are known as microplastics. Microplastics can be categorized into two classifications—primary and secondary microplastics. The main categories of microplastics include molded plastic powders, surface blast cleaning scrubbers, industrial plastic nanoparticles, and microbeads commonly present in cosmetic items. Moreover, spherical or cylindrical virgin resin pellets are widely used in plastic production processes. Secondary microplastics are generated by the breakdown or fragmentation of bigger plastic trash (*Foo et al., 2022*). Microplastics can present a significant danger to ecosystems and marine species due to their widespread distribution and the possibility of being consumed by marine organisms (*Li et al., 2019*).

There are several ways aquatic organisms can be exposed to plastic materials, including entanglement, ingestion, and interaction. Entanglement refers to the entrapment of the organism, including ghost fishing (*Law, 2017*). Ingestion of plastic debris could be intentional, accidental, or indirect (*via* ingesting organisms that have ingested the plastic). It has been observed in various animals, from planktonic invertebrates to large aquatic mammals (*Al-Thawadi, 2020*). The negative impacts of microplastics on aquatic animals have been identified to suppress growth performance, hinder reproductive functions, potentially induce neurotoxicity, depress feeding and foraging activity, alter oxidative stress, destroy metabolic responses (*Jimoh et al., 2023*), and alternatively increase mortality rates among aquatic organisms with an accumulation as low as 184 µg/L (*Li et al., 2021*). Microplastics accumulated in the neural system are also known to interfere with the defense mechanisms of aquatic animals, thereby acting as highly high-risk stressors by decreasing phagocytic activity (*Mallik et al., 2021*) and disrupting the lysosomal membrane (*Sharifinia et al., 2020*). The impacts of microplastic pollution extend to human health, the economy, tourism, and beach aesthetics (*Joesidawati, 2018*). Furthermore, microplastics can act as vectors for metal pollutants. Therefore, it is urgent to develop microplastic monitoring programs for marine ecosystems' different components, such as water, sediment, and biota (*Xu et al., 2020*; *Welden, 2020*).

A monitoring program for environmental quality using bioindicators is crucial to assess the toxic effects on an organism. The bioindicator species selected for such a program should have a wide distribution, be abundant and tolerant to environmental conditions, and be easily sampled (*Xu et al., 2020*). Macrozoobenthos are advantageous as bioindicators due to their sedentary habitat and their sensitivity to changes in the ecological conditions of aquatic environments (*Sidik, Dewiyanti & Octavina, 2016*). There are two compartments from which benthic species can directly take up microplastics, depending on the animal's feeding behavior: the sediment and the water column. Epifaunal filter feeders, for example, will uptake microplastics suspended in the water above the sediment, while infaunal deposit feeders will uptake microplastics within the sediment. Also, microplastics deposited on the sediment can be resuspended by mechanical forces and become available to the water column and its turbulent processes at different scales (*Pinheiro, Sul & Costa, 2020*). Because organisms inhabiting and feeding in benthic habitats are at the base of food

webs, it is crucial to understand how microplastics interact with and affect them (*Wright, Thompson & Galloway, 2013*). The main form of interaction is the uptake and assimilation of particles in the digestive system (*i.e.,* ingestion). However, benthic species also interact with microplastics in other ways, which could affect the movement of these pollutants within benthic environments. They may bury, mobilize, and even act as stepping-stones for the transmission of microplastics to other environmental compartments such as the nekton (*Pinheiro, Sul & Costa, 2020*).

The macrozoobenthos community consists of five groups: mollusca, polychaeta, crustaceans, echinoderms, and other groups of several small taxons such as Sipunculidae and Pogonophora (*Ndale, Restu & Wijayanti, 2021*). Barnacles (Crustaceans: Cirripedia) are a group of crustaceans that dominate certain zones of rocky coastal areas and are considered excellent biomonitoring organisms due to their widespread and abundant distribution around the world, easy collectibility, sessile adult life, and relative tolerance to contaminants (*Vaezzadeh et al., 2021*). Barnacles can potentially serve as bioindicators for microplastics because they are sessile and can accumulate microplastics from surrounding waters, thus aiding in identifying microplastic sources (*Xu et al., 2020*). According to the research (*Scotti et al., 2023*), provide an assessment of the efficiency of the biofouler Lepas (*Lepas*) *anatifera* Linnaeus, 1758 in capturing microplastics and microfiber particles floating in the water column. In this context, pelagic gooseneck barnacles are collected at fixed moorings in the Capo Milazzo Marine Protected Area (MPA). Fibers and fragments were found in the digestive tract of 30% of the 120 specimens collected. The ingested debris was mainly fibers (85.9%) of synthetic (30.6%) and natural (11.7%) origin, with lengths ranging between 1 and two mm (33.3%) and transparent (47.2%) (*Scotti et al., 2023*). Another investigation conducted in Hong Kong examined the presence of microplastics in four species of barnacles over 30 locations. The study found that the median quantity of microplastics varied from 0 to 8.63 particles per gram of wet weight, or 0 to 1.9 individual particles per barnacle. Among the several types of microplastics, fibers were found to be the most prevalent (*Xu et al., 2020*).

Surabaya produces significant plastic waste, estimated at 400 tons, which could result in significant environmental problems if not adequately managed (*Ni'am et al., 2019*). The East Coast of Surabaya is particularly susceptible to plastic pollution due to human activities in the coastal areas, fisheries, and tourism, as well as the discharge of plastic waste from rivers (*Ni'am et al., 2019*). The research area is located in the Suramadu Bridge Area. The Suramadu Bridge crosses the Madura Strait. Kenjeran Beach is a coastal area that directly faces the Madura Strait. On land, it is mainly characterized by tourism activities, fishermen's houses, and mangrove ecosystems. The waters of Kenjeran Beach are regularly used for fishing and marine tourism activities (*Cordova, Purwiyanto & Suteja, 2019*). A study conducted by *Kurniawan & Imron (2019)* revealed that a quantity of visible plastic debris (VPD) gathered along the seashore of the Madura Strait in both the dry and wet seasons. Polyethylene terephthalate accounted for the majority of plastic types distributed, reaching a share of 59.77% during the wet season. During the dry season, low-density polyethylene was the primary plastic component collected in all sampling points, making up 73.13% of the collected VPD composition. To address this pressing issue, a comprehensive

study is necessary to understand the characteristics and composition of microplastics in this region. Furthermore, identifying effective biomonitoring organisms for microplastic pollution is critical. Barnacles have shown promise as bioindicator species for microplastics since they are sessile and can accumulate plastic particles from surrounding waters, making microplastic sources identifiable (*Xu et al., 2020*). Therefore, this study aims to explore the potential of barnacles as a bioindicator species for microplastic pollutants and to characterize the visual and polymers of microplastics in barnacles in the East Coast Waters of Surabaya. The findings of this study will contribute to the development of monitoring programs and help mitigate the impact of microplastics in the marine environment.

## MATERIALS & METHODS

### Experimental design

Barnacle sampling was carried out in 2021 at the Suramadu Bridge Area on the East Coast of Surabaya, with coordinate points of 112.7799857 BT and −7.1905940 LS (Fig. 1). The sampling location was chosen because abundant barnacle species were found and fishing activities under bridges could be one of the causes of microplastics. The analysis of microplastic size, shape, and color on the barnacle samples was performed at the Ecology Laboratory, Department of Biology, Faculty of Science and Data Analytics, Institut Teknologi Sepuluh Nopember. The analysis of the microplastic polymer composition in barnacle, water, and sediment samples was collected at the Material Characterization Laboratory, Department of Materials Engineering and Metallurgy, Faculty of Industrial Technology and Systems Engineering, Institut Teknologi Sepuluh Nopember.

### Barnacle, water, and sediment sample collection

Fifty individual barnacle samples were collected from the research site in March 2021. The barnacles were removed from a pole of the Suramadu Bridge in the intertidal zone and placed in sealed glass bottles to ensure the durability of ingested microplastics. The bottles were stored in a cool box (*Xu et al., 2020*). The identification of barnacle samples is based on morphological characteristics of the hard part of the shell and shell cover plate, using Darwin's identification book from 1851 and 1954, as well as *Chan, Prabowo & Lee (2009)*. The barnacles obtained at the research site were identified as the *Amphibalanus amphitrite*. The barnacles underwent morphometric length, width, and tissue weight measurements. The barnacles used as research samples were in the size range of 0.4−0.7 cm in length and 0.2−0.6 cm in width. Water sampling was conducted by pulling a plankton net for 10 min at 2 knots, with a net size of 80 $\mu$m. The sample was taken horizontally at the surface at the study site, and the net was rinsed using water from the direction of the mouth of the net towards the cod end. The cod end was then released slowly, and the water sample in the cod end was rinsed using 70% ethanol to keep the water samples in good condition. All water samples were collected in sample bottles, sealed tightly, and placed in a cool box (*Prata et al., 2019*; *Zhang et al., 2020*; *Viršek et al., 2016*). The sediment samples were collected using an Ekman bottom grab (Model LEG-150 PROTIRTA standard size, sample area 232 cm$^2$) from the top 5 cm depth. Samples of 400 grams were collected at the study site, and the coordinates were marked using GPS. The sediment samples were placed in previously

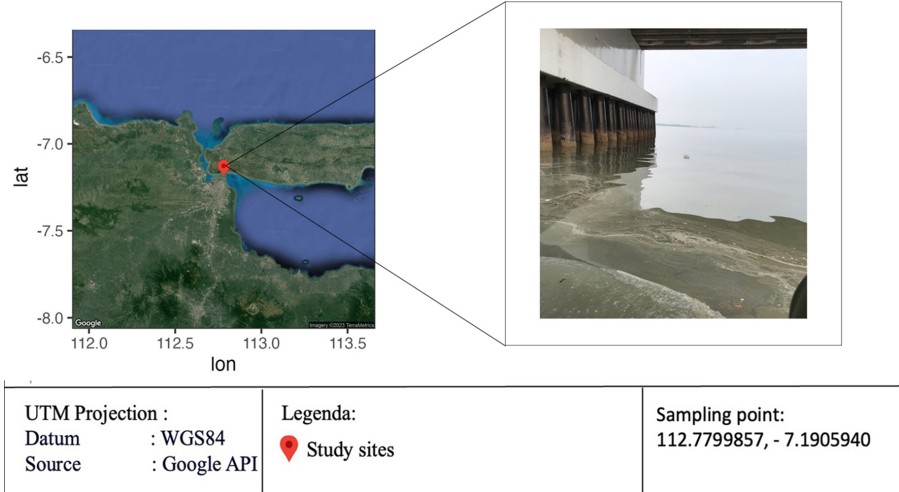

**Figure 1** Sampling location on Suramadu Bridge Pillar, East Coast of Surabaya (adapted from Google Maps, 2023). Map imagery credit: ©2023 TerraMetrics.

labeled plastic bags and stored in a cool box for further analysis (*Wang & Wang, 2018*; *Prata et al., 2019*).

## Sample preparation for microplastic analysis

To prevent air contamination, the trial table is sterilized with 75% ethanol before measurements of the length and width of the barnacle shell are taken using a funnel term. The wet weight of the barnacle soft tissue is then separated from the body using forceps and measured to the nearest 0.01 g. Five barnacles were put together as one replicated and placed in a 100 ml beaker glass. Next, 60 ml of KOH 10% (*Rochman et al., 2015*) is added to each replicate, and the solution is incubated at 40 °C for 48 h. The solution is then filtered through Whatman paper no. 41 (pore size 20 μm) and placed in a petri dish for further analysis (*Xu et al., 2020*). For water, samples are transferred into a test tube and dried in an oven at 80 °C for 24 h. To degrade organic materials, 2 ml of 30% $H_2O_2$ is added to each sample (*Masura et al., 2015*). The remaining water samples containing solid matter are mixed with 30% $H_2O_2$ at a volume ratio of 1:1 and left to stand for 24 h. The sample is closed using aluminum foil and left for 24 h before incubation in a water bath at 80 °C until the solution is clear. The clear solution is then dried on Whatman filter paper no. 41 using a vacuum pump (*Gewert et al., 2017*). For microplastic analysis in sediment samples, sediment samples are dried at 74 °C in an oven for 24 h. Then, 200 g of sediment is mixed with 600 ml of concentrated NaCl in a beaker glass and stirred for 2 min before being left for 1 h until the sediment settles. This process is repeated twice, and the supernatant containing microplastic particles is carefully transferred to another beaker glass (*Cordova, Hadi & Prayudha, 2018*). The supernatant is filtered through Whatman filter paper no. 41 using a vacuum pump, and the filter paper is placed in a petri dish for further analysis (*Xu et al., 2020*).

## Analysis of visual characteristics of microplastics

Visual analysis of microplastics is based on their morphological characteristics, including shape, size, and color. All microplastic particles were observed, photographed, and marked under an Olympus® SZ61 stereo microscope with the Optilab Advance 2.2 tool using Optilab Viewer software for visual analysis. The Optilab Advance 2.2 cameras, installed with ocular lenses on stereo microscopes (USB Digital Microscope 1600X Zoom Magnifier Monocular Lens), are connected to laptops for real-time observation. The Raster Image software is used to measure microplastics based on the most extended microplastic scales. Microplastic particles were categorized into four size classes: class 1 (1–10 µm); class 2 (10–50 µm); class 3 (50–100 µm); and class 4 (>100 µm) (*Shen et al., 2021*). Microplastic color analysis can vary greatly, leading to subjectivity in visual recognition. Therefore, the standard size and color sorting system (SCS System) is used to categorize plastic pieces effectively based on size and shape. The SCS system classifies microplastic shapes into five types: fiber, film, fragments, pellets, and foam (*Crawford & Quinn, 2017*).

## Analysis of the polymers composition of microplastics

The chemical composition of microplastics was analyzed using Attenuated Total Reflection Fourier Transform Infrared Spectroscopy (ATR-FTIR) Thermo Scientific Nicolet 1S10. The spectra were investigated and subtracted from the baseline in the scan range of 400–4000 $cm^{-1}$ at a resolution of two $cm^{-1}$. ATR-FTIR allows the analysis of large-sized microplastics (>500 µm) and also enables the direct analysis of microplastic particles with irregular shapes without preparation (*Wang & Wang, 2018*). There were ten replications of the barnacle samples; we only used the replications with the highest number of microplastics, namely replications 1, 2, 4, 8, and 10, for FTIR testing. Next, we also tested water and sediment samples. Of 155 particles suspected of microplastics in barnacles, 97 were observed for polymer analysis. For water and sediment samples, 17 particles and 24 particles suspected of being microplastics were observed for polymer analysis. Using the instrument's software (OMNIC Picta v1.7, Thermo Fisher Scientific Inc., US), the examined particle spectra were compared to a number of built-in reference spectra libraries. The libraries that were utilized were the HR Polymer Additives and Plasticizers (PAP), HR Hummel Polymer and Additives (HPA), Hummel Polymer Sample Library (HPSL), HR Aldrich Polymers (AP), and HR Spectra Polymers and Plasticizers by ATR–corrected (SPPATRc). Matches against the ATR corrected ATR libraries were preferred for determining the polymer identification (Copyright, 2008 Thermo Fisher Scientific Inc. for Nicolet FT-IR, Thermo Fisher Scientific Inc., Waltham, MA, USA). Additives, plasticizers, and comonomers such as acrylonitrile and propylene glycol that are used to create polymers have been counted as microplastics (*Carrillo-Barragán et al., 2024*).

## Quality control

To ensure quality control, all materials were washed with distilled water and ethanol to eliminate any cross-contamination. Additionally, the work surface was disinfected using ethanol. In addition, distilled water and NaCl solutions used for MPs separation in sediment samples were also analyzed for contamination and identified as being free of

MPs. The results revealed that both solutions were found to be devoid of any microplastics. Furthermore, the KOH solutions utilized for barnacle extraction and the $H_2O_2$ solutions for water extraction were examined for contamination and determined to be devoid of microplastics. Analysis showed that neither of the solutions contained any microplastics. Before usage, the filters were examined under a microscope to verify their absence of microplastic particles. The samples were consistently handled using latex gloves and stainless-steel tweezers (*Flores-Ocampo & Armstrong-Altrin, 2023*). Upon acquiring the microplastics, they were appropriately preserved in sealed glass petri dishes. To avoid contamination from clothing, the researcher stood at the edge or a distance from the sample while keeping and collecting it, wearing only cotton coats. Cotton laboratory coats were also always worn to avoid microplastic fibers from clothes in the laboratory air. In order to reduce the accumulation of microplastics in the laboratory, we aimed to examine the samples in a controlled setting with little air circulation (such as closed doors and windows and limited human activity within the laboratory) (*Shruti et al., 2022*).

## Data analysis

The data were analyzed to determine if there were significant differences between microplastics in barnacles, water, and sediment. Before the test, the data were analyzed normally using the Shapiro–Wilk test. When the data is distributed normally, the Pearson Correlation test is performed. All statistical tests were conducted at a significant level of 5% ($a = 0.05$). Pearson correlation of barnacle species was used to analyze the correlation between the size of microplastics and the size of barnacles. All statistical analysis was done using SPSS (Armonk, NY, USA).

## RESULTS

### Barnacles on the East Coast of Surabaya

Only one barnacle species, A. *amphitrite*, was found at the research site because this species has a significantly higher resistance to disease. The hard shells of A. *amphitrite* protect the soft parts of the barnacle against predators and adverse conditions. The relatively higher resistance to physical stress may give A. *amphitrite* a competitive advantage over other more fragile species (*Encarnação & Calado, 2018*). A. *amphitrite* is found living attached to rigid natural substrates in the bridge pillars (see Fig. 2). The vertical density of barnacles found in this study is thought to be related to their harmful phototactic properties, which means that barnacles tend to stay away from light, resulting in a higher population of barnacles in the water column than at the water's surface (*Nasution & Mudzni, 2016*). The morphology of A. *amphitrite* is characterized by a six-sided shell that appears to be pursed or condensed. The body's exterior is smooth and white with a vertical purple pattern, and there are no horizontal striations (see Fig. 3).

### Microplastic in barnacles, water, and sediment in the East Coast of Surabaya

This study found that all barnacles, water, and sediment samples contained microplastics. Microplastics were found in the barnacle samples, with 155 particles and an average of 15.5
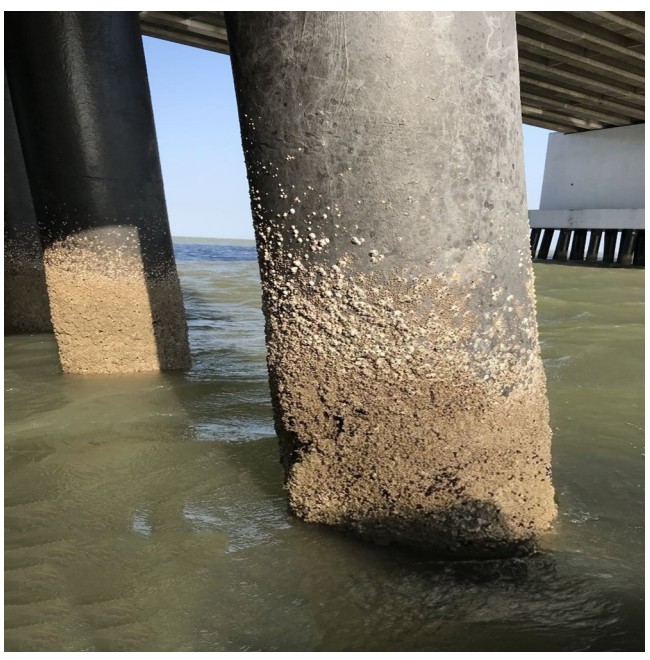

**Figure 2** Appearance of pillars showing substrate attachment of barnacles.

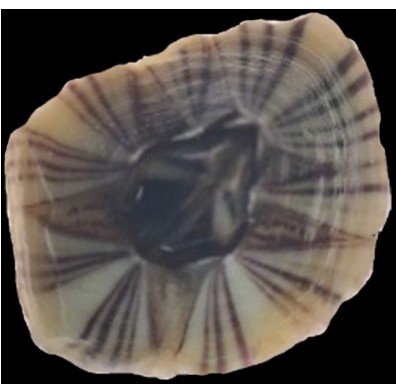

**Figure 3** Morphology of *Amphibalanus amphitrite* found at the site (magnification of 8×).

± 5 particles per replication. The average abundance of microplastics in barnacles was 5.37 particles per gram (see Table 1).

Water and sediment samples contained microplastics, with as many as 17 and 24 particles, respectively. The total abundance of microplastics in the waters off the East Coast of Surabaya was 0.38 particles/m³, while the total abundance in sediments was 60 particles/kg.

## Microplastic size

The most commonly found size class of microplastics was class 1, with 79 particles identified (78 in barnacles, 1 in water). Class 2 had 61 particles (45 in barnacles, nine

**Table 1  The number of microplastics found in barnacles (*Amphibalanus amphitrite*), water, and sediments in the East Coast of Surabaya.**

|  | Number of microplastics (particles) | Microplastic abundance |
|---|---|---|
| Barnacles | 155 | 5.37 particles/g |
| Water | 17 | 0.38 particles/m$^3$ |
| Sediments | 24 | 60 particles/kg |
| Total | 196 | |

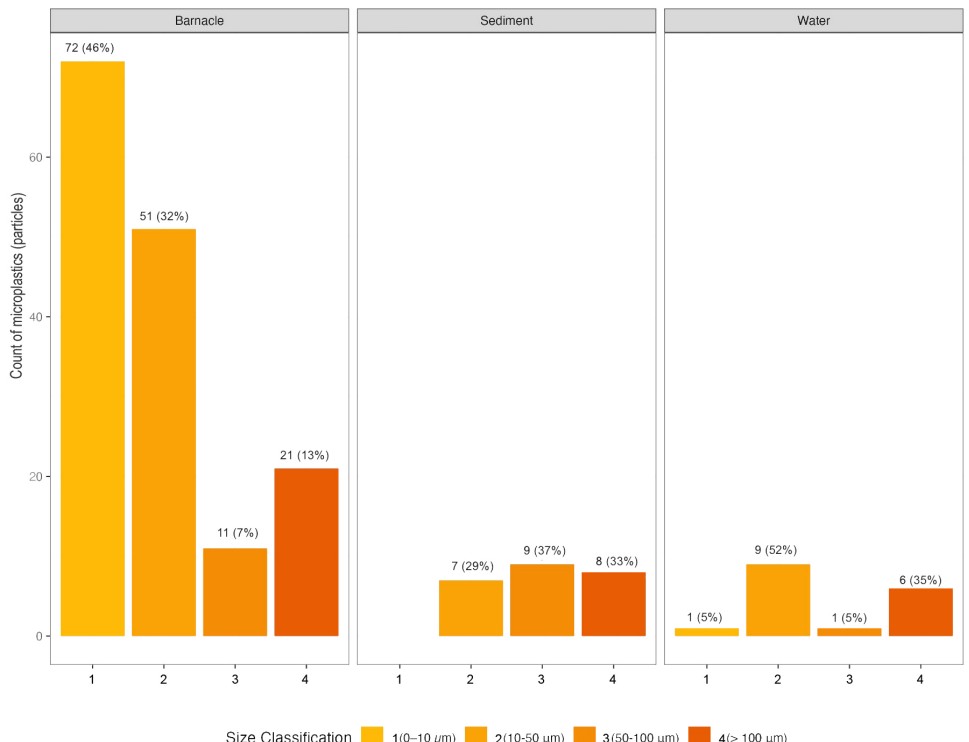

**Figure 4  Microplastic size in barnacles, water, and sediments in East Coast of Surabaya.**

in water, and seven in sediment), class 4 had 35 particles (21 in barnacles, six in water, and eight in sediment), and class 3 had 21 particles (11 in barnacles, one in water, and nine in sediment) (See Fig. 4). The correlation test results between barnacle length and microplastic length showed a negative correlation value or were inversely proportional to r = −0.411 ($p < 0.05$), categorized as having a strong enough correlation.

## Microplastic shapes

Visual observations of microplastics in barnacles, water, and sediments along the East Coast of Surabaya revealed the presence of three types of microplastics: fragments, fibers, and pellets. In barnacles, fragments (104 particles), fibers (47 particles), and pellets (four particles) were the dominant shapes of microplastics. In water and sediments, most

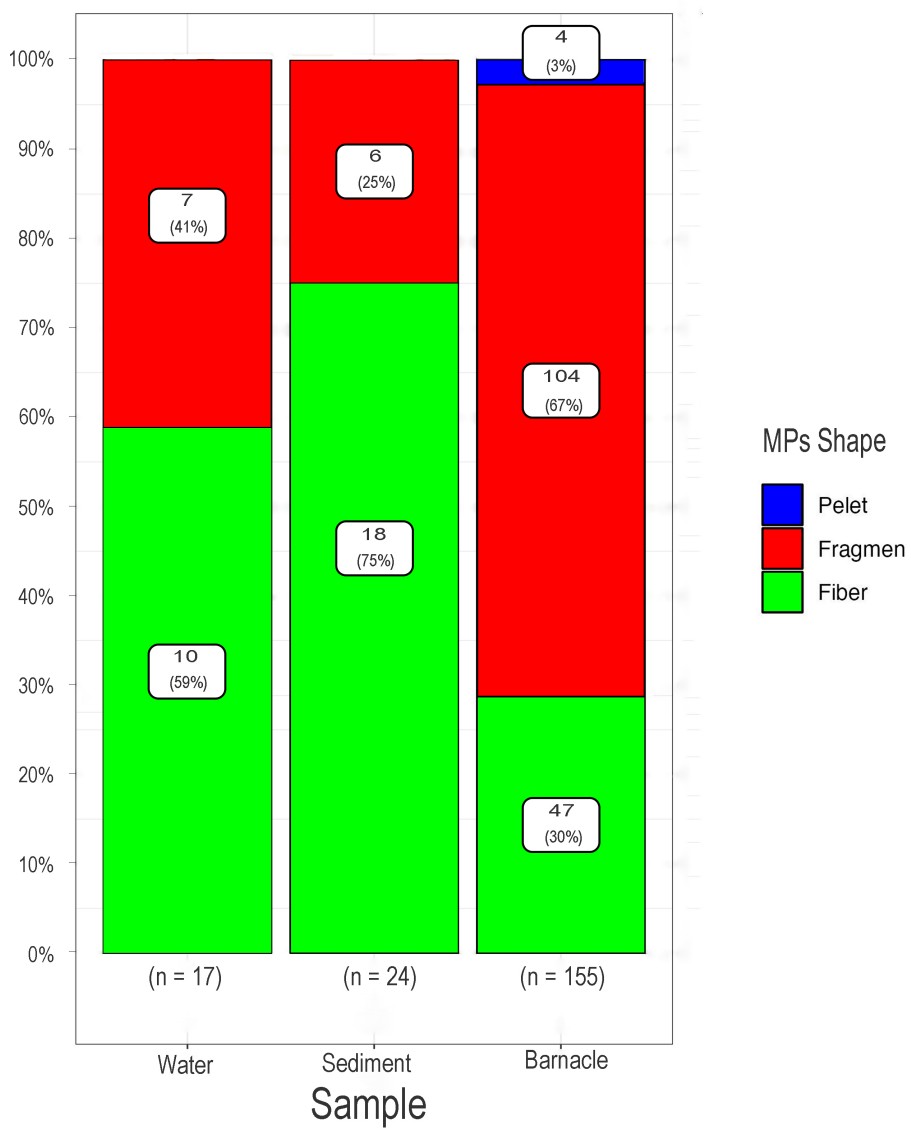

**Figure 5** Percentage of microplastic shapes in barnacles, water, and sediments found on the East Coast of Surabaya.

microplastics were in the shape of fibers (10 particles and 18 particles), followed by fragments (seven particles and six particles), with no pellets found (See Fig. 5).

## Microplastic color

This study identified six different colors of microplastics: blue, red, yellow, black, brown, and green. In the barnacle samples, blue was the dominant color (69.7%), followed by brown (12.9%), black (10.9%), red (5.1%), and yellow (1.3%). In the water samples, blue was also the dominant color (41.2%), followed by brown (23.5%), red (11.8%), black (11.8%), and green (11.8%). Sediment samples were primarily blue (91.6%), with

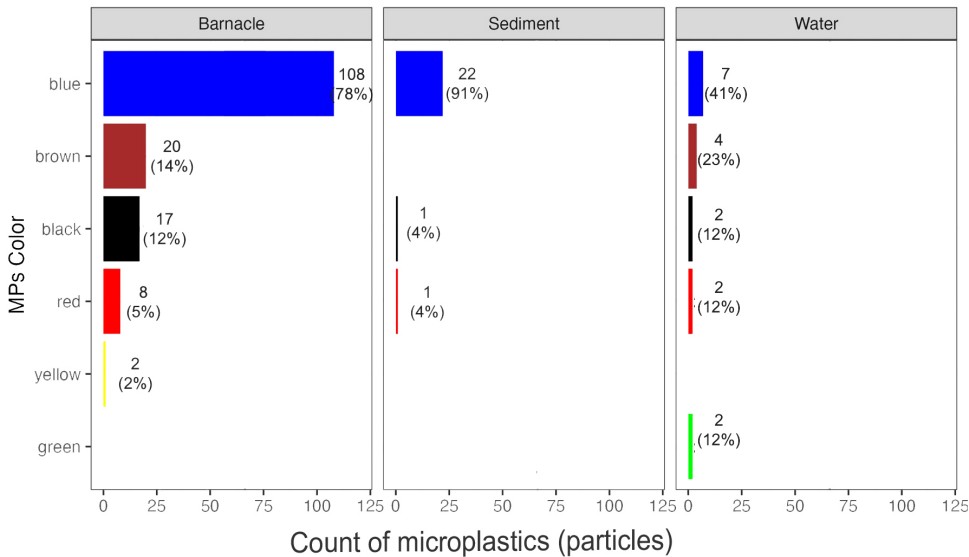

**Figure 6  Percentage of microplastic color in barnacles, water, and sediments found on the East Coast of Surabaya.**

smaller amounts of red (4.2%) and black (4.2%). Figure 6 provides a detailed graph of the percentage and color of microplastics.

## Microplastic polymer types

Attenuated Total Reflection Fourier Transform Infrared Spectroscopy (ATR-FTIR) was used to analyze barnacle, water, and sediment samples for microplastic particles. The ATR-FTIR analysis of barnacle samples revealed the presence of four different microplastic polymers consisting of cellophane, poly (1,3-oxathiolane), hydroxyethylcellulose, and hexa methylol melamine. Meanwhile, cellophane, Pullulan P2000, Pullulan P800, and polyester with kaolin filler were found for polymer analysis of water samples. No plastic polymers were found in the sediment samples. Cellophane accumulation in this study was 36%, followed by poly (1,3-oxathiolane) at 29%. Figure 7 provides a spectrum of microplastic that found in barnacle and sediment.

## DISCUSSION

### Microplastic in barnacles, water, and sediment in the East Coast of Surabaya

This is likely due to the pollution of the East Coast of Surabaya by various types of garbage, mainly plastic waste. Plastic waste dominates about one-third of the debris found in the waters (*Anjarwati et al., 2017*). Microplastics are small particles, typically micrometers in size, that result from the breakdown of larger plastic materials and fibers released during laundry and cleaning (*Anjarwati et al., 2017*). This assertion is supported by the condition of the research site, where plastic waste is observed floating on the water's surface, and a significant amount of plastic waste is present on the coast (see Fig. 8).
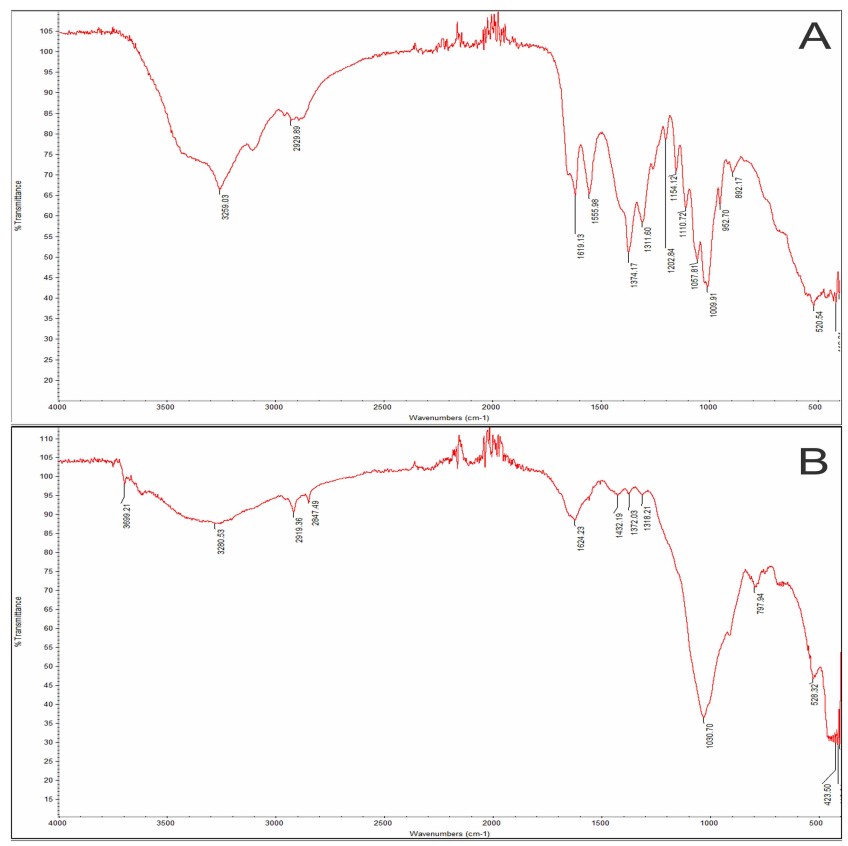

**Figure 7** **Attenuated Total Reflection Fourier Transform Infrared Spectroscopy (ATR-FTIR) spectra and match degrees of selected microplastics.** (A) Cellophane in barnacles and (B) cellophane in water.

Barnacles, being filter feeders, catch food by sticking out their cirri to capture food particles (*Xu et al., 2020*). Previous studies have shown that adult barnacles can digest microplastics (*Browne et al., 2008*). Other field studies conducted in three different sites in Thailand have found 0.23−0.43 particles per gram of tissue in *Amphibalanus amphitrite* (*Thushari et al., 2017*). Microplastics originate from the fragmentation of larger plastics transported by river run-offs, tides, and wind are sourced from the ocean, including fishing gear, aquaculture equipment, and clothing fibers from household waste. The high concentration of microplastics is also attributed to the passing ships that significantly contribute to microplastic pollution (*Ayuningtyas, 2019*). The amount of microplastics found at the bottom of the sediment was greater than on the water's surface, which is affected by factors such as gravity, currents, waves, and density. Microplastics settle in sediment when the water density is lower than that of microplastics (*Laksono, Suprijanto & Ridlo, 2021*). Microplastics with lower densities in the water will float in the water column, while those with higher densities will sink and settle in the sediment (*Victoria, 2017*). Regarding the abundance of microplastics in the water, this study found levels of 0.38 particles/m$^3$. Natural factors that affect the abundance of microplastics in the water include currents and waves, as the magnitude of waves can cause stirring and lift debris from the

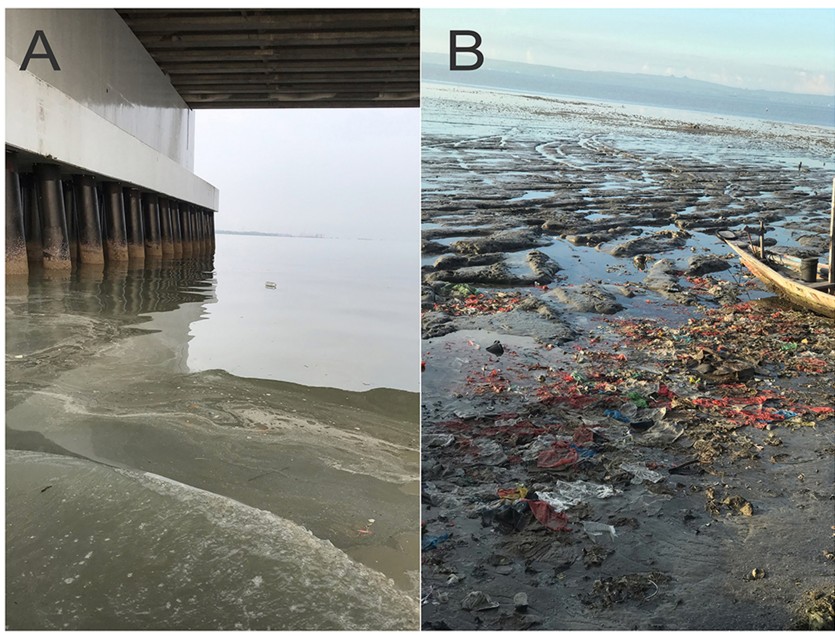

**Figure 8** **The condition of the research site.** (A) appears to be a plastic bottle floating under the Suramadu bridge and (B) pieces of crepe plastic on Kenjeran Beach.

bottom of the water to accumulate on the surface (*Muchlissin et al., 2020*). The movement of waste particles occurs due to the deflection of currents and waves in the ocean, with the weight and amount of microplastics determining their trajectory. Lighter microplastics are carried away by currents and tides (*Febriani, Amin & Fauzi, 2020*). Anthropogenic activities around the waters are another factor causing microplastic pollution. For instance, microplastics can originate from domestic activities such as washing clothes or from agricultural activities where wastewater from farms is often untreated and becomes a source of pollutants that empties into the ocean (*Permatasari & Radityaningrum, 2020*).

## Microplastic size, shapes, and color

The size range of microplastics is significant because it determines their potential impact on biota and ecosystems (*Nor & Obbard, 2014*). Microplastics below 40 µm have the same size range as most microplankton and nanoplankton and have a greater potential to be digested by various organisms. Generally, the size limit of microplastics is operationally defined by sampling and analytical methods (*Zhang et al., 2020*). A negative correlation can be interpreted as the larger the barnacle, the smaller the microplastics. This is thought to occur due to the process of thickening food in barnacles. Larger barnacles can be selective about food to recognize the presence of microplastics. However, this argument is inconsistent with the research of *Xu et al. (2020)*, which stated that the size of microplastics has a very weak correlation with barnacle size for species A. *amphitrite*: $r = 20.117$, $p = 0.485$, $n = 38$.

This study indicates that fragments are the most common shape of microplastics found in barnacles (67%). Fragments are often textured and dense and likely originate from the

fragmentation of plastic packaging, including drink bottles, plastic bags, and PVC pipes (*Victoria, 2017*). Plastic waste can also influence the prevalence of fragmentary shapes due to tourist and household activities, such as plastic bottles, mica packaging, and other objects with high plastic content (*Azizah, Ridlo & Suryono, 2020*). In water and sediment, the dominant shape of microplastics was fiber (58% and 75%). The long, colored fibers found in marine habitats may come from domestic waste, including synthetic cloth, fishing boat waste, and fishing gear such as fishing nets and lines (*Ayuningtyas, 2019*). Fishing activity is a significant contributor to fiber microplastics, as most fishing nets are made of fiber (*Victoria, 2017*; *Azizah, Ridlo & Suryono, 2020*). The study's findings suggest that the dominant shape of microplastic particles (fragments and fibers) likely originates from secondary sources, such as plastic waste that has degraded in the marine environment over time (*Nor & Obbard, 2014*). Pellets (2.5%) are another shape of microplastic found in barnacles, which are round-shaped and primarily derived from plastic factories, cleaning and beauty products, and resin powders (*Victoria, 2017*). Pellets are often produced directly by raw material manufacturers making plastic products, including those used to shape larger plastic products, and micro-scale products, such as industrial plastics and medical devices, including those used in pharmaceutical applications. The results of this study support the findings of *Viršek et al. (2016)*, which state that the most commonly found microplastic particles in water ecosystems are fragments from larger plastic waste and fibers from textile fibers. The shapes of microplastics discovered in this study are consistent with those reported by *Xu et al. (2020)*, who found microplastic fibers (95.7%), fragments (3.4%), and pellets (0.8%) in barnacles. This study revealed the absence of other forms of microplastics apart from fibers and fragments in the water samples. This is caused by the use of techniques carried out by researchers, namely by carrying out a drying process on water samples at a temperature of 80 degrees. This was confirmed by *Munno et al. (2018)*, who confirmed that the choice of chemical digestion technique to remove natural organic matter and isolate microplastic particles, as well as the specific conditions under which the method is applied, can have a significant impact on the recovery of certain types of microplastics particle. The investigation revealed that heating conditions impacted two of the five categories of microbeads present in the personal care products evaluated. Beads melt when the solution's temperature, resulting from the application of heat or an exothermic oxidation process, exceeds 60 °C (*Munno et al., 2018*).

Some studies suggest that certain microplastics, such as fibers, are commonly colored red and blue and primarily originate from domestic water waste, such as laundry water, from residential areas and water waste management plants. If microplastics are still highly concentrated in color, it suggests that they have not undergone significant discoloration (*Dekiff et al., 2014*). A yellowish color indicates prolonged exposure to the sea and oxidation. Chocolate color is caused by the oxidation of polymers in microplastics that have been exposed to sunlight for an extended period. The black color of microplastics suggests they are composed of polystyrene (PS) or polypropylene (PP) and may also contain pollutants such as polycyclic aromatic hydrocarbons (PAHs) and absorbed polychlorinated biphenyls (PCBs) (*Azizah, Ridlo & Suryono, 2020*; *Laksono, Suprijanto & Ridlo, 2021*).

## Microplastic polymer types

However, no microplastic polymers were detected in sediment samples using FTIR analysis, likely due to the challenges associated with analyzing irregularly shaped fragments, which is usually a refractive error and interference (*Dris et al., 2018*). Other studies have shown that the degradation can cause changes in the FTIR spectrum, interfering with the identification process (*Jung et al., 2018*). The dominance of cellophane (CP) in this study is consistent with previous findings by *Xu et al. (2020)*, who identified CP as one of the three dominant polymers in barnacles, with polyethylene terephthalate (PET) and polypropylene (PP). *Ding et al. (2018)* also reported the common presence of CP and PP in marine bivalves. Cellophane was the most commonly identified microplastic fibers and fragments in this study. Although marketed as "biodegradable", the low biodegradability of regenerated cellulose, such as cellophane (36%), suggests that biodegradable plastics are a "false solution" to marine waste. Cellophane is extensively used in food packaging and as a release agent in fiberglass rubber production. The dominance of cellophane as a microplastic type has also been reported at several sampling points in the United Kingdom estuarine complex and salt lakes in China (*Zhang et al., 2019*). Poly (1-3-oxathiolane) (29%) was found in this study, as well as in zooplankton, according to *Sun et al. (2018)*. The prevalence of fiber and cellulose in commercial species highlights their limited ability to recognize and avoid microplastics during consumption (*Wu, Zhang & Xiong, 2018*). Rayon, a regenerated cellulose commonly found in personal hygiene products and textiles, is likely a primary source of cellulose in marine environments (*Acharya et al., 2021*). Hydroxyethyl cellulose (7%) is a water-soluble, non-ionic polymer used in various industrial fields, such as color thickening, textile finishing, and thickening in cement mortars (*Abdel-Halim, 2014*). Hexa methylol melamine (7%) contains resins used in the production of coatings and plastics for cans, wires, and cars (*Dsikowitzky & Schwarzbauer, 2015*). Pullulan P200 (7%) and Pullulan P800 (4%) have high potential in thin filmmaking, are transparent, odorless, tasteless, and consumable, and are widely used as packaging materials and to produce dental strips (*Kraśniewska, Pobiega & Gniewosz, 2019*).

## Barnacles as potential microplastic bioindicators

Barnacle crustaceans, known for their high tolerance to environmental pressures, are commonly used in coastal areas for monitoring marine pollution (*Scotti et al., 2023*). Barnacles feed by extending their cirri to trap brine shrimps, copepods, and phytoplankton from the water column (*Xu et al., 2020*). Due to their feeding efficiency, surface position in the water column, and opportunistic behavior, these organisms are susceptible to consuming microplastics. As a result, they can be regarded as appropriate species for evaluating the presence and dispersion of microplastics and fibers in the surrounding waters (*Scotti et al., 2023*). The previous study by *Goldstein & Goodwin (2013)* reported that 385 individuals of gooseneck barnacles (*Lepas* spp.) collected from the North Pacific Subtropical Gyre (NPSG) had plastic particles present in their gastrointestinal tract, ranging from one plastic particle to a maximum of 30 particles. Results by *Thushari et al. (2017)* show that barnacles are highly susceptible to microplastic accumulation showing the highest micro debris density compared to other filter feeders. Barnacles use a filter-feeding
mechanism by extending their spines outward to create water currents. During their feeding, water with suspended particles, food materials, and plastic waste enters the body cavity without selection and accumulates in their bodies.

The *Amphibalanus amphitrite* of barnacles found in this study holds promise as a potential microplastic bioindicator due to its cosmopolitan distribution and abundance in diverse marine habitats worldwide. A. *amphitrite* primarily engages in regular and rapid planktivory, with the addition of simultaneous maxilliped microfiltration. During normal beat, water flow through the mantle cavity is intensified to enhance the filtration of microscopic food. On the other hand, during a fast beat, there is a higher probability of capturing particulate food through the large cirri (*Pasternak & Achituv, 2007*). A previous study by *Xu et al. (2020)* supports this, noting the presence of A. *amphitrite* in various habitats, including mudflats, piers, and dams. The high abundance of microplastics found in barnacles compared to surrounding sediments and seawater (5.37 particles/g *vs.* 0.06 particles/g and 0.38 particles/m$^3$, respectively) indicates that barnacles can track microplastic exposure in the environment. Furthermore, the smaller size of A. *amphitrite* reduces the efficiency of microplastic expenditure from the digestive tract, which can result in higher microplastic content in the habitat. This study also revealed similarities in the characteristics of microplastics found in barnacles, water, and sediment, including the dominance of fragment and fiber shapes of microplastics, as well as the blue color of the particles. The analysis of microplastic polymers found in barnacles and the environment also showed a high percentage of cellophane (36%), which is consistent with the results of the previous study by *Xu et al. (2020)* that found cellophane (57.69%), polyethylene terephthalate (PET, 11.54%), and polypropylene (PP, 9.62%). To strengthen the findings of this study, it is crucial to increase the sample size and replicate measurements across various environmental compartments. This would provide a more comprehensive understanding of microplastic distribution and accumulation patterns. Furthermore, incorporating multiple bioindicator species can validate observed correlations, and conducting long-term monitoring studies can reveal temporal trends in microplastic concentrations, enhancing the assessment of microplastic pollution in marine ecosystems.

## CONCLUSIONS

This study identified 196 microplastic particles in *Amphibalanus amphitrite*, water, and sediments. Barnacles had the highest concentration of microplastics (155 particles), followed by sediment (24 particles) and water (17 particles). The size, shape, and color of the microplastics varied across the samples, with barnacles dominated by class 1 size (1–10 µm), fragments, and blue color. Water and sediment were dominated by class 2 and 3 sizes (10–50 µm and 50–100 µm, respectively) and fiber shapes, with blue being the most common color. Cellophane was the most dominant chemical composition found in all samples. Based on the findings, *Amphibalanus amphitrite* shows promise as a potential bioindicator for microplastics, particularly the cellophane type. Although we did not calculate correlations due to the lack of sample replication across environmental compartments, our observations suggest a linear relationship between its population and the accumulation of this specific type of microplastic.

## ACKNOWLEDGEMENTS

We thank the lecturers from the Faculty of Biology at Universitas Jenderal Soedirman for knowledge sharing regarding this research.

### Funding

Miftakhul Sefti Raufanda received a scholarship from the Indonesia Endowment Fund for Education (LPDP), under the supervision of the Ministry of Finance of the Republic of Indonesia. LPDP also supports facilitating the publication of this article. This work was supported by the Department of Biology FSAD ITS and the Directorate of Research and Community Service while facilitated the research activities through Research Contract No. 1882/PKS/2022 Batch 2. The external funders had a role in the decision to publish. The funders had no role in study design, data collection and analysis, or preparation of the manuscript.

### Grant Disclosures

The following grant information was disclosed by the authors:
the supervision of the Ministry of Finance of the Republic of Indonesia.
Indonesia Endowment Fund for Education (LPDP) under the supervision of the Ministry of Finance of the Republic of Indonesia.
The Department of Biology FSAD ITS and the Directorate of Research and Community Service: 1882/PKS/2022.

### Competing Interests

The authors declare there are no competing interests.

### Author Contributions

- Miftakhul Sefti Raufanda conceived and designed the experiments, performed the experiments, analyzed the data, prepared figures and/or tables, authored or reviewed drafts of the article, and approved the final draft.
- Aunurohim Aunurohim conceived and designed the experiments, analyzed the data, authored or reviewed drafts of the article, and approved the final draft.
- Romanus Edy Prabowo conceived and designed the experiments, analyzed the data, authored or reviewed drafts of the article, and approved the final draft.

### Data Availability

The raw data is available in the Supplemental File.

### Supplemental Information

Supplemental information for this article can be found online at http://dx.doi.org/10.7717/peerj.17548#supplemental-information.

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
