# Peer review of "Barnacle analysis as a microplastic pollution bioindicator on the East Coast of Surabaya"

_PeerJ, doi:10.7717/peerj.17548_

## Round 0.1 · original submission · Major Revisions

Please incorporate all comments of reviewers and please submit revisions along with a point-to-point rebuttal letter. You may ignore or include the citations suggested by reviewers, at your discretion.

Reviewer 1 ·

Basic reporting

In this study, the predominant species Amphibalanus amphitrite was evaluated as a bioindicator for the detection of microplastic pollution. It is explained by the authors that it shows a linear correlation, especially with Cellophane type microplastics.

In the introduction, the importance of the subject, the study area and the status of Barnacles are explained. However, it is recommended to elaborate with current references. It seems that this important topic needs further deepening.

Similarly, I recommend that the discussion section be rewritten in detail.

Experimental design

In my opinion, information about the sampling site should be given in the material methods, so that it would be healthier to describe the environment with the results of water sediment contamination.

Validity of the findings

I recommend that the reason for choosing the area shown in Figure 1 as the sampling site should be explained.
I recommend that given the FTIR Spectrum of microplastics in specimens, water, and sediment samples as figure.

Additional comments

1. Introduction, explain the toxic effects of marine microplastics using current literature.
2. The relationship between marine benthic invertebratres and microplastics should be explained in more detail.
3. In my opinion, information about the sampling site should be given in the material methods, so that it would be healthier to describe the environment with the results of water sediment contamination.
4. I recommend that the reason for choosing the area shown in Figure 1 as the sampling site should be explained.
5. Line 98; please show it as Figure 1.
6. I recommend that given the FTIR Spectrum of microplastics in specimens, water, and sediment samples as figure.
7. Discussion section between lines 268 and 292; should be written more clearly with references. The phrase "various organisms" should be improved by explaining it with references.
8. Line 239; Pitriana et al., 2020;
Line 242; Fatmasari, 2016;
Line 280; Laksono, Suprijanto and Ridlo, 2021 were not found in the reference list.
Line 316; Mohamed Nor and Obbard, 2014, Please check the author's name.
Line 474; The reference was not cited in the manuscript.
9. Lines 303 and 491; Please write the species name in italics.

Reviewer 2 ·

Basic reporting

This article is an important source of information on microplastic pollution on the east coast of Surabaya. Thus, the results and discussion contained is of interest to the scientific community as well as to possible legislators concerned with the environment and nature conservation. Is for this reason that, I believe, the scientific English presented is not enough for a worldwide public to understand the information described.
The english language improvements are needed in different secton of the paper:
Lines 17 and 19 (East coast, not coastal), 189, 235, 244 (desease instead of desorder?) , repeated words(lines 27,28,29 and 31,80), or ideas/phrases (from 49 to 51).

Figures are self-explanatory although sometimes the choice made for results presentation was not entirely suitable (flat pie chart for a big number of variables).

In discussion, some references should be included on lines 275,279,299,306,311

Experimental design

The material and methods of the article is excellent. It will be easy to replicate the findings for those interest in.
I just missed an explanation or justification for sampling between March and June and not an annual sampling where we could see a difference in the results. Is this sampling choice relevant for the results? Could we see other outcomes if you sampled in other seasons? Is barnacles feeding behaviour subject to seasonal variation?
All these questions do not invalidate the results, but their answer could resteate them.

Validity of the findings

Findings of this research are strong and will add valueable information for the general knowledge of microplastic pollution in the study area

Reviewer 3 ·

Basic reporting

Clear, unambiguous, professional English language used throughout.
It is commendable how well the manuscript reads given that the authors are not native English speakers, however, there are numerous grammatical errors throughout the entire document, including the title. For example, “the East Coastal of Surabaya…” should be, “…the East Coast of Surabaya.”

Intro & background to show context. Literature well referenced & relevant.
1) The introduction could benefit from more information about the variety of MP shapes that occur. The authors only mention that microplastics break down into fragments (line 45) which although accurate, does not alert the reader to the fact that fragments are only one possible shape of MPs in the environment. Furthermore, the description of the formation of microplastics fails to include other methods of formation, specifically physical abrasion with other particles (line 45), although this is contradicted later in the discussion (lines 310-316).
2) There is little justification for using barnacles as bioindicators of MPs. The only mention of MPs and bioindicators occurs in lines 70-75 and there is no direct mention that previous studies observed barnacles ingesting MPs.
3) The introduction to plastic waste in Surabaya (Lines 77-93) lacks quantifiable data on how much contamination occurs in the region. Adding this information would enhance the transition from general background on using bioindicators with MPs to specifically targeting the use of barnacles as bioindicators in Surabaya.
4) Lines 91-93 discusses the implications of the study and how it will, “...mitigate the impacts of microplastics on the marine environment” but there is little mention of these impacts, making it difficult to understand how the findings will benefit society. The authors should include more information about the effects of MP ingestion on organisms, which organisms ingest plastics, and the considerable debate about whether organisms can be used as indicators of microplastic contamination.

Structure conforms to PeerJ standards, discipline norm, or improved for clarity.
1) The correct section headings are present, however, the information in each section does not necessarily belong there. For example, there is a significant amount of introduction material repeated in the discussion, and there is information in the discussion that belongs in the introduction. Furthermore, there are many results in the discussion that should not be there.
2) The sections are unbalanced. That the Introduction is only two pages, and the Discussion is more than 5 pages indicates that there is information in the discussion that should be in the introduction.
3) Information about how correlations were interpreted does not merit a table.
4) The authors should refrain from referencing figures in the discussion. That is another indication that there are too many results in the discussion section.

Figures are relevant, high quality, well labelled & described.
1) Figure 1: does not show the study site in a context that is understandable to people in other countries. A separate panel showing the relative position of the sampling location within Surabaya would be useful. In addition, another image depicting the actual posts where barnacles were collected from the sampling site would help readers better understand the sampling design. There is a mistype under the figure: it should read “Legend” rather than “Legenda”.
2) The y-axis on Figure 2 should be more descriptive to aid the reader in identifying which MP abundance information is displayed. The size classes listed in the legend do not correspond to the definition of each size class in-text. Lines 196-197 list classes as either, “1, 2, 3, 4” whereas the legend lists size classes as, “a, b, c, d”.
3) Figure 3: x- and y-axes should be switched so the different compartments (water, sediment, barnacle) are on the x-axis. There is currently no x-axis label, and the y-axis label should be more informative - it is unclear whether “sample” refers to a replicate or environmental compartment. Also, color-blind people cannot differentiate between red and green so a different color palette should be selected. The title of the Figure should also be changed from “...Form” to “...Shape” to better reflect the terms used in the manuscript text and microplastics literature. There is no figure 3b showing a barnacle as indicated in the text (Line 178).
4) Figure 4: The dotted line needs to be defined because it is unclear what it is showing. It is also unclear what (n Particle) means.
5) Figure 5: legend should be titled “Polymers” rather than “Molecules” to better reflect the terminology most often used to describe MP materials in the literature.
6) Figures 6, 7, and 8 were all missing from the provided files and were assumed to not have originally been submitted, despite multiple references to these figures several times in the discussion section.

Raw data supplied (see PeerJ policy).
1) All datasheets should have a more informative naming scheme. It is difficult to understand what each sheet contains, especially the sheets labeled, “R”, “homogen”, “morfometri”, and “Mps teritip”. There are also translation issues, such as “teritip” carried throughout several sheets and the headers for columns “A – E” on sheet “Mps teritip” is also mistranslated.
2) Metadata describing the process of how some of the values were calculated would be beneficial. For example, in the sheet labeled “FTIR” there are percentages listed and all values seem to be based on a total of 23 plastic particles , whereas the results suggest 97 particles were analyzed with FTIR. There are also several sets of data in columns O-R of sheet “Mps in sediment and water” that do not have labels.
3) Spectral data are missing. The authors should provide reference and example spectra for the observed shapes to verify the ATR-FTIR identification techniques.
4) It is difficult to understand how many plastics were identified. There were 196 putative plastic particles, of which 97 were supposedly analyzed with ATR-FTIR (line 97). However, the raw data show that 23 particles were confirmed as “plastics”. Despite only 23 confirmed MPs, subsequent figures and analyses are based on the initial 196 particles, which may overestimate the number of MPs found in the samples.

Experimental design

Original primary research within Scope of the journal.
Yes.

Research question well defined, relevant & meaningful. It is stated how the research fills an identified knowledge gap.
The research question was not well-defined. The authors pose two aims:
1) to explore the potential of barnacles as a bioindicator species for microplastic pollutants (vague and difficult to assess based on the current study)
2) to characterize the visual and chemical properties of microplastics in barnacles in the East Coastal Waters of Surabaya (able to be accomplished from the current study)
The knowledge gap stated is based on lack of information in coastal Surabaya.

Rigorous investigation performed to a high technical & ethical standard.
Microplastic investigation techniques have advanced considerably in the last few years. The present study includes some of these more advanced techniques (e.g., ATR-FTIR), however, the microplastic sampling and analysis do not appear to have been conducted using appropriate QA/QC protocols, and contamination was not accounted for. There do not appear to be any violations of ethics standards.

Methods described with sufficient detail & information to replicate.
1) The first paragraph of the results section (lines 175-180) should be incorporated in the methods section. This information would replace the description of identifying mussels in lines 109-110. The mention of, “...for each species” insinuates multiple species of barnacles were collected and analyzed, which was not the focus of the study (or what occurred according to the data presented).
2) The reported plankton tow speed and duration is helpful (Lines 112-117), but more information is needed to explain where these tows occurred and when to replicate the study. It is also difficult to understand how many tows were conducted at each site and whether multiple sites were visited. This information is needed to address sample size and whether the sampling conducted characterizes the area or was only from one spot.
3) Please explain why ethanol was used to rinse the samples into the cod end bottle.
4) More information is needed for the sediment sampling process. The procedure lacks details regarding collection locations and what type of grab was used to harvest the 400 g samples. Citations are needed to justify the information in lines 118-119 that states “MPs are most abundant in the highest tide zone”.
5) More information on how barnacles were processed is needed. It seems that each replicate consisted of 5 barnacles composited together, but it could also be interpreted to mean every 5th barnacle was considered a separate replicate.
6) Brand and instrument names should be included for many of the components/tools described. This includes the plankton nets, filter papers (both pore size and material), sediment grabber, and ATR-FTIR microscope.
7) There are serious methodological concerns regarding preparation of the water samples.
a. It is stated that samples were dried in an oven at 80°C for 24 hours. However, Munno et al. (2018) observed that some plastic polymers melt at temperatures exceeding 70°C and although not all plastic polymers melt, it is important to at least discuss the possibility that the observed plastics may not accurately reflect those found in the environment.
b. The plankton tows were only digested with a single dose of hydrogen peroxide over 24 h, which was justified with Masura et al. (2015). However, recent revisions to this protocol require that additions of peroxide continue until bubbling no longer occurs when the sample encounters the H2O2 (Hurley et al., 2018).
c. The methodology described in the current study also includes heating in a water bath at 80°C during the 24 h peroxide digestion. Applying heat during the digestion can accelerate the reaction and increase temperatures above 80°C, potentially exacerbating the plastic-melting issue. Therefore, the plankton tow samples may not be comparable to other studies and limitations of the results must be discussed for this work to be valid.
8) The sediment samples were not chemically digested, and the only processing was a density separation step. No citations were provided in lines 139-143 where the sediment processing was described and should be added to justify the choice of methodology and improve repeatability.
9) More information is needed to replicate the microscopy techniques using both ATR-FTIR and the Optilab Advances 2.2 tool. For the ATR-FTIR, the type of instrument and settings must be included. The type of microscope and total magnification used to analyze MP shape, color, and size is also needed.
10) The first sentence of the data analysis section states that differences in MP concentrations between sediment, water, and barnacles were assessed (lines 163-164), although there were no statistical tests comparing the concentrations across any of the environmental compartments. The only statistics consisted of one correlation between barnacle size and microplastic abundance, and it was based on 10 data points.
11) There is very little quality analysis/quality control described in the methods. The only instance of controlling for MP contamination occurs in lines 126-127, where the authors state they prevented air borne contamination by wiping surfaces with ethanol prior to processing. However, MPs are continuously suspended in the air and deposition occurs continuously throughout the processing procedure (from air, researcher’s clothing, etc.), which may have introduced MPs in the time between the wiping of benchtops and quantifying MP contamination. Therefore, it is difficult to trust that all particles observed in either of the three compartments (barnacles, sediment, water) were exclusively from the field and not contaminant MPs that deposited during the processing steps.

Validity of the findings

Impact and novelty not assessed. Meaningful replication encouraged where rationale & benefit to literature is clearly stated.
The only mention to the benefit to literature/our knowledge of MPs occurs briefly in the introduction (lines 89-93) and the discussion (lines 374-383). However, more information is needed throughout the discussion to help the readers better understand how this study improves our understanding of using invertebrates as MP indicators, particularly when this is a debated issue in the literature (none of this literature was included).

All underlying data have been provided; they are robust, statistically sound, & controlled.
The data are provided as required, however, there are discrepancies in numbers of particles for each analysis compared to what is reported in the text (see #2 under raw data supplied) and the sample size of barnacles is extremely low. Barnacles occur in dense aggregations and are relatively easy to collect. It is unclear whether the barnacles were distributed over a large area or from one bridge support. The small sample size from (likely) one location is particularly concerning when trying to determine whether barnacles might be acceptable indicators of microplastic contamination. There is no “control/clean” location for comparison, and information about whether barnacles take up microplastic in proportion to availability is needed.

Conclusions are well stated, linked to original research question & limited to supporting results.
1) The authors suggest the use of barnacles as bioindicators for MP contamination (lines 380-383), but this is not necessarily supported by the results. This study found that barnacles ingest MPs in the Surabaya region and there was an inverse correlation between body size and MP abundance. However, samples seem to only have been harvested from one site, failing to capture whether the abundance of MPs in the barnacles is reflective of different MP concentrations in the sediment and water. Therefore, the use of barnacles as bioindicators of MPs was not truly assessed in this study.
2) Furthermore, the discussion fails to mention the use of invertebrate bioindicators in other studies/regions and the debate surrounding use of organisms as indicators of MP contamination in general (see the following articles for more in-depth discussions: Li et al., 2019; Ward et al., 2019; Ding et al., 2021).
3) The discussion section includes various findings from previous studies but frequently fails to provide appropriate citations. The following lines require citations: 260, 262, 269, 271, 285, 290, 295, 306, 312, and 355.

Suggested Literature:
Ding, J., Sun, C., He, C., Li, J., Ju, P., & Li, F. (2021). Microplastics in four bivalve species and basis for using bivalves as bioindicators of microplastic pollution. The Science of the Total Environment., 782. https://doi.org/10.1016/j.scitotenv.2021.146830

Hurley, R. R., Lusher, A. L., Olsen, M., & Nizzetto, L. (2018). Validation of a method for extracting microplastics from complex, organic-rich, environmental matrices. Environmental science & technology, 52(13), 7409-7417.

Li, J., Lusher, A.L., Rotchell, J.M., Deudero, S., Turra, A., Bråte, I.L.N., Sun, C., Shahadat Hossain, M., Li, Q., Kolandhasamy, P., Shi, H. (2019). Using mussels as a global bioindicator of coastal microplastic pollution. Environ. Pollut. 244, 522–533. https:// doi.org/10.1016/j.envpol.2018.10.032.

Munno, K., Helm, P. A., Jackson, D. A., Rochman, C., & Sims, A. (2018). Impacts of temperature and selected chemical digestion methods on microplastic particles. Environmental toxicology and chemistry, 37(1), 91-98.

Ward, J. E., Zhao, S., Holohan, B. A., Mladinich, K. M., Griffin, T. W., Wozniak, J., & Shumway, S. E. (2019). Selective ingestion and egestion of plastic particles by the blue mussel (Mytilus edulis) and eastern oyster (Crassostrea virginica): implications for using bivalves as bioindicators of microplastic pollution. Environmental science & technology, 53(15), 8776-8784.

---

## Round 0.2 · Minor Revisions

Please incorporate all remaining minor suggestions of reviewer and submit with a point-to-point response letter.

Reviewer 1 ·

Basic reporting

This manuscript is based on the evaluation of A. amphitrite (Barnacles), an important benthic organism for the marine ecosystem, as a bioindicator of plastic pollution. A. amphitrite, water and sediment samples were collected from sampling sites heavily exposed to plastic pollution on the east coast of Surabaya. The types of plastic were determined by ATR-FTIR analysis. The correlation between plastic types and samples was evaluated. The population of A. amphitrite was found to be positively correlated with the accumulation of cellophane (CP).

In my opinion, this revised manuscript has undergone major changes and improvements according to the reviewers' recommendations. The Introduction and Discussion sections have been rewritten and references to microplastic pollution and organism exposure have been given.

Experimental design

I think that the section on material methods and design is well explained in such a way that the aims of the research are clear and this study is suitable for PeerJ.

Validity of the findings

After the major revision, the presentation and comparison of the results have been improved, I think it is appropriate.

Additional comments

There are some deficiencies in the writing of references and cited in the text. I recommend that these should be corrected.

1. Line 45; Liu et al. 2021 was not found in the references list.
2. Line 61; please check the reference date (Jimoh et al., 2023).
3. Line 63; please indicate whether the reference is 2021a or b; or correct it in the reference list (Li et al. 2021). Because lines 590 and 594; Li et al. 2021a and 2021b were not found.
4. Line 117; please check the reference date (Cordova et al., 2019).
5. Line 336; please check the reference date (Xu et al., 2023).
6. Line 417; please check the reference date (Jung et al., 2017).
7. Line 429; please check the reference date (Wu et al., 2019).
8. Line 446; please write species in italics (Lepas spp.)

---

## Round 0.3 · accepted · Accept

The paper has been addressed all comments and improved. Thus, it is accepted.